# Occupational Noise: Auditory and Non-Auditory Consequences

**DOI:** 10.3390/ijerph17238963

**Published:** 2020-12-02

**Authors:** Adam Sheppard, Massimo Ralli, Antonio Gilardi, Richard Salvi

**Affiliations:** 1Department of Communicative Disorders and Sciences and Center for Hearing and Deafness, University at Buffalo, Buffalo, NY 14221, USA; salvi@buffalo.edu; 2Department of Sense Organs, Sapienza University of Rome, 00185 Rome, Italy; massimo.ralli@uniroma1.it (M.R.); antonio.gilardi@uniroma1.it (A.G.)

**Keywords:** occupational noise exposure, ultra-high frequency, otoacoustic emissions, hidden hearing loss, tinnitus, hyperacusis, dementia, sound pressure level, continuous equivalent level

## Abstract

Occupational noise exposure accounts for approximately 16% of all disabling hearing losses, but the true value and societal costs may be grossly underestimated because current regulations only identify hearing impairments in the workplace if exposures result in audiometric threshold shifts within a limited frequency region. Research over the past several decades indicates that occupational noise exposures can cause other serious auditory deficits such as tinnitus, hyperacusis, extended high-frequency hearing loss, and poor speech perception in noise. Beyond the audiogram, there is growing awareness that hearing loss is a significant risk factor for other debilitating and potentially life-threatening disorders such as cardiovascular disease and dementia. This review discusses some of the shortcomings and limitations of current noise regulations in the United States and Europe.

## 1. Introduction

The World Health Organization (WHO) estimates that ~16% of all disabling hearing loss stems from occupational noise exposure [1]. However, this number underestimates the global health burden of occupational noise because the metrics used to compute occupational noise-induced hearing loss fail to consider other potentially debilitating hearing-related impairments such as tinnitus, hyperacusis, and deteriorated ability to understand speech in noise. Moreover, current regulations do not take into account widespread co-morbidities associated with noise-induced hearing loss (NIHL) such as cognitive decline, social isolation, increased stress, anxiety, depression, and higher risk of cardiovascular disease [2,3,4,5]. 

Decades of research has led to the development of guidelines designed to protect individuals from hazardous occupational noise exposure [6,7,8]. Current noise regulations are largely based upon research studies that identified noise conditions (e.g., duration, intensity, peak pressure, spectrum, and intermittency) that resulted in audiometric threshold shifts within frequencies responsible for speech understanding. New research suggests this approach may underestimate the impact of low-intensity, long-duration noise exposures [9,10,11,12,13], fails to consider auditory deficits that occur without concomitant hearing loss [14,15,16], and overlooks non-auditory health consequences of noise-induced hearing loss [17,18,19,20]. In the following sections, we briefly review current occupational noise regulations and discuss additional auditory and non-auditory consequences of occupational noise that negatively impact quality of life and long-term health.

## 2. Occupational Noise Regulations

Some of the prominent governmental bodies that have established regulations/recommendations to avoid hazardous noise exposure include the Occupational Safety and Health Administration [7], the National Institutes of Occupational Safety and Health [8], and the European Union (EU) [21]. The current noise regulations aim to reduce the risk of hearing loss at frequencies important for speech perception, but do not address other auditory and non-auditory deficits.

Contemporary regulations consider four major interacting variables when determining if a noise dose is hazardous. First, and most obvious, is the noise sound pressure level (SPL), represented in the logarithmic unit of decibels (dB). Second is the spectral content; noise levels are most commonly measured using the dB(A) scale, which attenuates sounds below 1 and above 8 kHz. This filtering is done because high-frequency noise (namely those centered around 4 kHz) pose a significant risk for auditory threshold shifts [22], although for extremely high intensity sounds, it is more common to measure sound intensity using the dB(C) scale, which has little sound attenuation except for frequencies below approximately 60 and above 8000 Hz. The third factor important for calculating noise dosages is the duration of exposure; the longer the noise exposure duration, the greater the risk of audiometric threshold shifts, at least until thresholds reach an asymptote at durations ≥ 24 h [23]. Hence, when determining if a noise exposure is hazardous, both the noise level and duration need to be evaluated simultaneously. A common method of integrating both the noise intensity and duration is the equivalent continuous sound level (L_eq_), which represents the energy mean of the noise level averaged over a defined period. When L_eq_ acoustic measurements are made for 8 h with the A-weighting filter, the term L_Aeq,8h_ is used to indicate the type of frequency weighting and the duration of the exposures. Finally, the fourth factor, which is mainly considered in environmental noise ordinances, is the time of day. Several studies have suggested that nighttime noise exposure increases the risk of hearing loss, annoyance, risk of cardiovascular disease, sleep disturbance, and cognitive impairment [17,24]. For environmental noise, the time of day when the noise exposure occurs is expressed as L_DEN_ (day/evening/night level), which indicates the average A-weighted sound pressure level over a 24-h period with an additional 10-dB penalty during nighttime hours (11:00 p.m.–7:00 a.m.) or a 5-dB penalty during evening hours (7:00 p.m.–11:00 p.m.) [18] (i.e., 10 dB or 5 dB is added to actual L_eq_ measure to reflect night or evening exposure, respectively). The combination of these factors is the basis for understanding current occupational and environmental noise regulations. 

### 2.1. OSHA

In the United States, OSHA, the regulatory body responsible for establishing guidelines to avoid hazardous occupational noise, defines the permissible exposure level (PEL) as 90 dB(A) for 8 h. The PEL is adjusted for exposure durations shorter or longer than 8 h using a 5-dB exchange rate. Accordingly, the PEL would be 95 dB(A) for an exposure duration of 4 h, whereas the PEL would be 85 dB(A) for a 16-h exposure [7]. Furthermore, OSHA limits the peak sound pressure level of noise exposure to 140 dB(C), independent of exposure duration. Although OSHA indicates 90 dB(A) as the PEL, employers are required to establish a hearing conservation program when the exposure level reaches 85 dB(A) [7]. OSHA requires conservation programs to include (1) annual audiometric assessments, (2) employee training on the risks associated with excessive noise exposure, and (3) hearing protective devices to be worn by employees [7]. Annual threshold assessments conduct pure tone thresholds at 0.5, 1, 2, 3, 4, 6, and 8 kHz using air conduction transducers. The annual thresholds are then compared to the worker’s baseline thresholds obtained at the time of employment to determine if a significant threshold shift occurred. Although frequencies are measured at 0.5, 1, 2, 3, 4, 6, and 8 kHz, a significant threshold shift is considered present only when thresholds increase an average of 10 dB or more at 2, 3, and 4 kHz in either ear. 

### 2.2. NIOSH

While OSHA is responsible for establishing regulations, the Occupational Safety and Health Act of 1970 (Public Law 91-596) tasked NIOSH with providing recommendations to avoid hazardous exposure to occupational noise. Importantly, these recommendations were meant to include occupational noise dosages that would reduce the risk of employees having “diminished health, functional capacity, or life expectancy as a result of his or her work experience” [8], health metrics that are not captured by basic threshold shifts. NIOSH first provided recommended exposure limits (REL) in 1972, and revised the RELs in 1998 [8]. Current recommendations are slightly stricter than those regulated by OSHA. NOISH’s REL defines 85 dB(A) for 8 h as the recommended exposure limit, and incorporates a 3-dB exchange rate. The 3-dB exchange is based on the equal energy principle, since an increase or decrease of 3 dB doubles or halves the total noise energy, respectively [25]. While this may not seem like a large difference, some studies indicate that deploying a 3-dB exchange rate in OSHA regulations would identify 1.5–3 times as many workers as being potentially at risk for NIHL [6,26]. In fact, the debate over the use of a 3-dB or 5-dB exchange rate has been ongoing for some time, and still continues today [27,28,29,30,31,32]. NIOSH also recommends an upper ceiling peak pressure limit of 140 dB(A) and recommends that a hearing conservation program be established for workers exposed to an 8-h time-weighted average (TWA) of 85 dB(A). The prevention program recommended by NIOSH is similar to what is regulated by OSHA; however, there is a notable difference in their criteria for determining what constitutes a noise-induced threshold shift. NIOSH recommends a standard threshold shift during annual threshold monitoring of 15 dB(A) at 0.5, 1, 2, 3, 4, or 6 kHz in either ear, spanning a much wider range of frequencies than OSHAs criteria [8]. 

### 2.3. EU

In the European Union, EU-OSHA regulates occupational noise based on the directive 2003/10/EC promulgated by the European Parliament [33]. The directive characterizes noise hazardousness at three distinct levels, which are referred to as the lower action level, the upper action level, and the noise limit level. Occupational environments are categorized into a lower action level if the L_Aeq,8h_ (called a daily noise exposure level and denoted in the directive as L_EX_) reaches 80 dB(A) or a peak pressure value of 135 dB(C). If noise reaches a L_Aeq, 8h_ of 85 dB(A) or has a peak sound pressure value of 137 dB(C), it is classified as an upper action limit level. Finally, the ceiling threshold for acceptable noise exposure is an L_Aeq, 8h_ of 87 dB(A) or a peak sound pressure limit of 140 dB(C), and there is no permissible exposure above this ceiling noise limit. The lower and upper action levels are distinct from each other in that they require increasing conservation practices, respectively. When noise reaches the lower action level, employers must make hearing protection devices available to employees and make annual hearing monitoring available; whereas if noise exceeds the upper action level, these practices stay in place and the employer must also take actions to actively reduce the noise level, either through engineering controls (e.g., noise attenuation) or organizational practices (e.g., reduced exposure time).

## 3. Environmental Noise Regulations

Recently, the WHO recommended guidelines for environmental noise exposures in the European Region [20]. It is recommended that: (1) Road traffic noise be limited to 53 L_DEN_ and 45 dB L_NIGHT_, (2) railway noise be limited to 54 L_DEN_ and 44 dB L_NIGHT_, (3) aircraft noise be limited to 45 L_DEN_ and 40 dB L_NIGHT_, (4) wind turbine noise be limited to 45 dB L_DEN_, and that (5) other leisure activities be limited to 70 dB L_Aeq_,_24 h_ [20]. In the U.S., environmental noise exposures are largely regulated by individual state and city ordinances and vary greatly by region. However, the U.S. Environmental Protection Agency (EPA) recommended that an individual not exceed a daily noise dosage of 70 dBL_Aeq, 24 h_, that community indoor levels be maintained below 45 dBL_eq_, and that certain community outdoor areas be maintained below 55 dBL_eq_ [34]. It is important to consider environmental noise exposure because it occurs alongside occupational noise exposures and may contribute to hearing loss. However, environmental noise is not accounted for when considering what occupational noise levels are considered hazardous to hearing. Therefore, individuals with high exposure to environmental noise at home or during leisure activities are at greater risk for hearing-related deficits if they are additionally exposed to high levels of occupational noise. 

The regulations outlined above were not designed to protect the entire working population from noise-induced hearing loss. Rather, they are meant to protect against a substantial portion of occupational NIHL while maintaining a feasible cost–benefit ratio.

For instance, a 40-year lifetime exposure to NIOSH’s REL still holds a risk of developing NIHL of ~8% and a 40-year lifetime exposure to OSHA’s PEL criteria still has a risk of NIHL of 25% [1,8]. Deploying comprehensive hearing conservation programs (i.e., protecting 100% of workers) may not be financially feasible [35]. Studies estimate that the cost of hearing conservation programs are on average ~$300–350/worker, and still yield a hearing impairment prevalence of ~15% [35,36]. Therefore, governmental regulations attempt to strike a balance between limiting hazardous noise exposure and maintaining economic growth and development. However, this balancing act could have unforeseen future consequences since current regulations only consider the impact of noise exposures on hearing thresholds and fail to consider several other noise-induced consequences, some of which come with a hefty financial burden to the healthcare system [37,38]. The consequences overlooked by contemporary occupational noise regulations can be broadly categorized into (1) auditory deficits beyond threshold shifts and (2) non-auditory health consequences as discussed below.

## 4. Significant Auditory Deficits Beyond Threshold Shift

### 4.1. Ultra-High Frequency Hearing Loss

The human cochlea can process a wide range of frequencies from ~20–20,000 Hz; however, standard clinical audiometric evaluations only sample hearing acuity from 250 to 8000 Hz, since they are putatively the most important for speech comprehension. The cochlea appears especially sensitive to noise-induced damage in the frequency regions between 3–6 kHz [39]. Consequently, the pure tone audiogram of patients exposed to hazardous levels of noise often take on a notched configuration centered around the 4-kHz frequency. For decades, the 4-kHz “noise notch” has been considered a hallmark of NIHL [22], and as such, the 4-kHz frequency region has been weighted more heavily when screening for audiometric shifts due to occupational noise [7]. However, recent studies suggest that frequencies >8 kHz may be more sensitive to noise-induced damage [40,41,42,43]. Riga et al. screened 151 noise-exposed workers’ hearing annually from 0.25 to 8 kHz and found that ultra-high frequency (12.5, 14, and 16 kHz) thresholds deteriorated the earliest, during the first decade of employment, and that elevated thresholds in the 2–4 kHz region were not observed until the second decade of employment [43]. In contrast, OSHA only assesses thresholds at 0.5, 1, 2, 3, 4, and 6 kHz and only flags a significant shift when thresholds worsen (i.e., increase) by an average of 10 dB 2, 3, and 4 kHz. NIOSH’s recommendations for noise monitoring are more sensitive, flagging a significant shift in thresholds when any 15 shift occurs at frequencies as high as 6 kHz, but still fails to consider noise-induced threshold shifts at the most sensitive frequencies (i.e., >10 kHz). 

### 4.2. Sub-Clinical OHC Dysfunction

The cochlea possesses a finely-tuned, biological electro-mechanical system that amplifies sound-induced pressure fluctuations. This inherent non-linear amplification of the cochlea lowers hearing thresholds and enhances frequency resolution, thereby improving hearing abilities in both quiet and noise [44]. Cochlear outer hair cells (OHCs) play a critical role in the non-linear amplification properties of the cochlea [45], but are also highly sensitive to noise-induced damage [46]. One assessment tool that is particularly sensitive to noise-induced OHC dysfunction is distortion product otoacoustic emissions (DPOAEs) which can be measured up to 16 kHz with some clinical instruments [47]. DPOAEs are generated by the electromotile response of OHCs, mediated by the motor protein prestin, and driven in part by the +80 mV endocochlear potential [45,48]. Contrary to common thought, DPOAEs are not tightly correlated to pure tone thresholds. In fact, it is relatively common for individuals with a history of noise exposure to have clinically normal audiometric thresholds (≤ 25 dB hearing level (HL), but show significantly lower (worse) DPOAE amplitudes [49,50,51,52,53,54]. 

Indeed, several studies have indicated that DPOAEs could serve as a more sensitive tool for identifying noise-induced damage in occupational settings [41,52,54,55,56,57]. For example, in a sample of 76 military personnel, those with prior noise exposure demonstrated significantly lower DPOAE amplitudes, even when thresholds were normal [54]. 

In addition, test re-test reliability in OAEs also demonstrates less variance than pure-tone thresholds [52,58]. In a sample of 38 subjects, OAE measures demonstrated test re-test variance of 2.9–3.1 dB (standard deviation), whereas pure tone thresholds had a variance of 4.9 dB [52], a critical difference when thresholds are tested in 5-dB steps and significant threshold shifts are defined by differences of 10 dB. Hence, OAE measures appear to be more sensitive at detecting early signs of noise-induced damage, indicating that OHC function has already deteriorated by the time pure-tone threshold shifts are detected during annual hearing conservation testing. 

### 4.3. Damage to IHCs and Afferent Synapses

Historically, OHCs were considered the cochlear structure most sensitive to noise-induced damage. However, recent animal studies now suggest that synaptic connections between inner hair cells (IHCs) and type I afferent nerve fibers could be more sensitive to noise damage than OHCs [59,60]. Immunohistochemical studies indicate that short-duration, high-intensity exposure to noise (~100 dB SPL, 2 h) yields selective damage to low-spontaneous rate high-threshold afferent nerve fibers in animals [61,62]. Interestingly, synapses for high-spontaneous rate fibers that are important for detecting low-intensity sounds (i.e., audiometric thresholds) are largely unaffected. Similar observations have been made in post-mortem human temporal bones, suggesting that cochlear synaptopathy likely also occurs in humans [63]; however, the functional consequences of synaptopathy in humans remain controversial [64,65,66]. In the context of occupational noise regulations, the threat of noise-induced cochlear synaptopathy occurring in humans seems somewhat low if regulations are strictly adhered to. Synaptopathic damage occurring in rodents is most reliably seen after exposures that would exceed the OSHA PEL (e.g., 2-h exposure would have maximum PEL of ~95 dB(A)) making such exposures impermissible without the use of hearing protection [67]. Nevertheless, this assessment could be flawed since evidence suggests that there is relatively poor compliance with the use of hearing protection in occupational settings (~20–40% non-compliant) [26,68,69]. As such, further research is needed to fully characterize the risk of occupational noise inducing cochlear synaptopathy in humans. 

### 4.4. Tinnitus and Hyperacusis

Tinnitus is a phantom auditory perception, described as a ringing, buzzing, or hissing sensation that can occur following excessive noise exposure [70]. Approximately 10–15% of adults experience tinnitus [71], and approximately 1% have debilitating tinnitus for which they seek medical treatment [72]. The prevalence of tinnitus is higher among adults exposed to occupational noise (~15%) compared to workers who report no occupational noise (~5%) [73]. Others have indicated the prevalence of tinnitus among noise exposed workers could be as high as 66% [55]. Similarly, noise-exposed combat veterans show a particularly high prevalence of tinnitus [37], in some cases as high as 50–70% [74,75]. Tinnitus not only has psychosocial consequences, but also financial ones for the Veteran Administration (VA), which lists tinnitus as the most prevalent and costly disability in the VA healthcare system. Recent estimates of VA disability payments for tinnitus totaled to a staggering $2 billion [76,77]. 

Hyperacusis, an abnormally low tolerance to moderate-intensity everyday sounds [78], is another auditory perceptual disorder that can present independent of hearing loss. Hyperacusis is highly comorbid with tinnitus, with approximately 86% of hyperacusis patients also reporting tinnitus [78,79]. Animal models suggest that excessive noise exposure can lower loudness discomfort levels (i.e., hyperacusis) [80,81,82]. However, there is a dearth of data linking hyperacusis to occupational noise exposure [83]. The most relevant studies of noise-induced hyperacusis come from those of professional musicians, which indicate hyperacusis has a prevalence of approximately ~18–28% when it occurs alongside other auditory disorders [84] and a prevalence of ~6% when it is the only reported hearing disorder [85]. Data from female pre-school teachers exposed to noise at work suggests a model of hyperacusis whereby the prevalence is increased by additional factors such as stress, annoyance, or unrelated leisure noise [86]. 

The exact pathophysiological mechanism(s) responsible for tinnitus and hyperacusis remains enigmatic. However, many researchers believe tinnitus emerges from excessive spontaneous neural activity [87,88,89,90] and hyperacusis results from enhanced sound-evoked neural activity [90] triggered by excessive/aberrant central gain control mechanisms [91,92,93]. Central gain functions in a homeostatic manner, attempting to centrally compensate for a reduced neural output peripherally (i.e., cochlea afferent fibers). Therefore, any of the previously discussed pathologies undetectable by threshold measures (i.e., ultra-high frequencies, sub-clinical OHC dysfunction, cochlear synaptopathy) could potentially be the underlying cause of tinnitus and/or hyperacusis. 

### 4.5. Hearing in Noise

Hearing in the presence of background noise is the most common hearing-related complaint and can occur in individuals with and without normal audiometric thresholds [14,94]. Difficulties hearing in noise, despite having normal audiometric thresholds, has been referred to over the decades as obscure auditory dysfunction [95], King Kopetzky syndrome [96], auditory processing disorder [97], and most recently hidden hearing loss [15]. Studies have yet to come to a conclusion on whether noise exposure directly results in hearing in noise difficulties [98,99]. On one hand, some have indicated that noise exposure results in poorer speech discrimination in background noise [15,100], while many others have failed to observe this association [64,101,102,103,104,105]. 

The pathophysiological mechanisms that make it difficult to discriminate in background noise are not fully understood; however, mounting evidence suggests it could be due to selective damage to IHCs and/or auditory nerve fibers. When chinchillas are treated with carboplatin, a commonly used anti-cancer drug, they exhibit a unique pattern of damage that selective destroys IHCs, while leaving OHCs intact. Despite this damage, animals display normal threshold acuity in quiet [16]. However, when thresholds are measured in the presence of background noise, chinchillas with selective IHC loss display significantly worse thresholds [106,107]. Moreover, one of the hallmarks of auditory neuropathy, a disorder where the pathology occurs within IHCs, afferent synapses, or spiral ganglion neurons, is difficulty hearing in noise [108,109]. Like the chinchilla model, these patients also have difficulty detecting tones in the presence of background noise [110]. 

Difficulty hearing in noise is commonly linked to the discussion of hidden hearing loss (a term some use synonymously with cochlear synaptopathy). If findings in animal models hold true in the translation to human studies, cochlear synaptopathy may provide an explanation for why noise exposure can impair hearing in noise. For example, evidence suggests that selective damage to IHCs and auditory nerve fibers does not affect thresholds, but does impair supra-threshold processing in noise [16,106,107]. In the interim, it is important to recognize other pathologies resulting from excessive noise that likely result in hearing in noise deficits. For instance, OHCs are also critically important for hearing in the presence of background noise [44], and their abnormal function can go undetected on standard audiometric assessments [52,54]. Furthermore, hearing loss occurring at ultra-high frequencies can also contribute to hearing in noise deficits [101,111,112,113,114], and may be more sensitive to noise damage than previously thought [43]. 

Taken together, several auditory deficits occurring from noise exposure can occur beyond the standard pure tone audiogram and are not considered in today’s hearing conservation programs. Outside of the auditory realm, recent studies also indicate several non-auditory consequences of excessive noise exposure, some of which are discussed in the following section.

## 5. Non-Auditory Health Consequences

### 5.1. Stress 

Unregulated stress responses can have large implications for numerous biological functions. The body’s stress response system attempts to maintain homeostasis through hypothalamic–pituitary–adrenal (HPA) axis negative feedback. In general, the paraventricular nucleus of the hypothalamus activates the HPA axis through the secretion of corticotropic releasing hormone (CRH), which provokes the release of adrenal corticotropic hormone (ACTH) from the anterior pituitary. ACTH then travels via the blood stream to the adrenal glands, which sit atop the kidneys, and activate the release of stress hormones such as cortisol, adrenaline, and noradrenaline When levels of stress hormones are sufficiently high they interact with the hypothalamus and hippocampus to shut off the HPA axis, resulting in negative feedback [115]. Chronic exposure to noise can be an indirect stressor capable of imposing a host of stress-induced scenarios [116]. For instance, chronic exposure to noise can result in sleep disturbances, increased difficulty communicating, and disrupted cognition [17], which in turn can disturb the HPA stress response system. 

Both occupational noise and chronic environmental noise can result in increased stress hormones such as cortisol, adrenaline, and noradrenaline [19,117]. Interestingly, changes in stress hormone levels are linked to the intensity and temporal aspects of the noise exposure. Intense, acute noise presented near the aural threshold of pain (130–140 dB SPL) results in increased release of cortisol [118], whereas acute noise presented at levels of 90–100 dB(A) increases the release of adrenaline and noradrenaline [119]. Furthermore, if the source of noise is novel to an individual’s environment (non-habituated), it is likely to result in increased release of adrenaline, whereas if the individual has been chronically exposed to the noise (habituated), the body is likely to display chronically elevated levels of noradrenaline [118]. Surprisingly, low levels of noise, far below the standards for protecting against hearing loss, can initiate the stress response. Increases in adrenaline and noradrenaline have been observed following environmental noise exposure as low as 60 dB_LAeq_ if the noise was disrupting to active processes (conversation, concentration, reactions, etc.). During sleep, traffic noise levels as low as 30 dB_LAeq_ have been shown to increase cortisol release [117]. In the context of occupational noise exposure, cortisol has shown a dose-dependent increase with increasing noise exposure levels [120]. The relationship between noise exposure and the body’s stress response is critical since prolonged levels of stress can increase the risk of life-threatening health conditions like cardiovascular disease [121,122]

### 5.2. Cardiovascular Disease

Cardiovascular disease (CVD) is an umbrella term used to refer to a host of disorders of the heart and blood vessels. CVDs are the number one cause of death globally, responsible for ~31% of all deaths [123], and pose an enormous fiscal burden on healthcare systems [124]. Chronic activation of the body’s stress response system is a recognized risk factor for the development of CVDs [125]. Since chronic exposure to noise effects the body’s stress response (discussed above), it is also thought to increase the risk of CVDs.

Both occupational and environmental noise exposure can increase the risk of CVDs. For example, in a large sample of industrially noise-exposed workers, the relative risk for developing coronary heart diseases was as high as 1.48, significantly higher than in age-matched, unexposed workers [126]. Similarly, individuals with higher levels of residential traffic noise are at greater risk for developing hypertension [127]. These results suggest that workers exposed to occupational noise in addition to residential traffic noise would be at even greater risk for CVD. However, it is difficult to make definitive associations between noise exposure and CVDs due to a large variety of confounding variables such as genetic differences, diet, history of smoking, alcohol use, physical activity, etc. Further research is needed to fully characterize the risk that noise exposure has on developing CVDs. 

### 5.3. Cognition

There is growing evidence to suggest that age-related hearing loss increases the risk of cognitive decline in older adults. This once provocative finding is supported by several meta-analytic reviews [128,129,130,131,132,133,134,135,136,137,138]. In fact, some have reported the association between hearing loss and dementia to be so strong that approximately 9.1% of all dementias could result from untreated hearing loss [131]. Like CVDs, the fiscal burden of dementia on today’s healthcare system is significant. In 2009, the cost of dementia-related healthcare was a staggering $422 billion [132].

Since it is extremely difficult to disentangle the contribution of past noise exposures on age-related hearing loss, it is plausible that noise could play a role in developing dementia. Indeed, animal studies support the notion that exposure to noise can disrupt cognitive functions such as learning and memory. Rats’ exposure to traumatic noise shows reduced neurogenesis in the hippocampus, a brain region critically important for spatial orientation and memory. Moreover, reduced neurogenesis is associated with deficits on learning and memory tasks [139,140,141]. Furthermore, even when animals are exposed to seemingly safe noise levels (75 dB_LAeq, 8 h_) for extended periods, they can display poorer performance on learning and memory tasks [142]. Similar to the theories of noise-induced CVDs, noise-induced consequences on learning, memory, and brain function could also be mediated by the altered stress responses, given that noise-induced hearing loss upregulates glucocorticoid receptors in the hippocampus, a critical component of the negative feedback network in the HPA axis [143]. 

## 6. Summary

Exposure to occupational noise poses a significant public health concern. It is well documented that exposure to noise above a critical level results in unrepairable sensorineural hearing loss. Noise regulations established for occupational settings do not protect 100% of workers from NIHL, but rather strike a balance between hearing conservation and economic development. The current occupational and environmental noise regulations may significantly underestimate the adverse health and societal cost of noise-induced hearing loss. There is growing awareness that noise exposures, even at moderate levels, may contribute to a host of other auditory and non-auditory consequences that can go undetected in annual hearing conservation monitoring programs. Taken together, these results suggest that the true fiscal consequences to our healthcare systems may be far greater than anticipated. Further research is needed to fully characterize the auditory and non-auditory consequences associated with occupational noise. Future research could uncloak (1) methods for earlier detection of auditory damage, (2) risks associated with combined noise exposure sources (i.e., leisure, environmental, and occupational) and (3) establish dose–response relationships between various noise conditions and non-auditory consequences.

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
