# Peer review of "Occupational Noise: Auditory and Non-Auditory Consequences"

_ijerph, 2020, doi:10.3390/ijerph17238963_

Round 1

Reviewer 1 Report

Review of Manuscript                                                                                      

Title: Occupational Noise: Auditory and Non-Auditory Consequences (Review)

Manuscript ID: ijerph-993583

Comments and suggested revisions:

Overall, this is a well written and well laid out manuscript. It is topical and of great relevance. Very minor revisions indicated below. Would like to see a few more references added to the Intro and to the section on DPOAEs and hearing protection usage. There are a few places where more explanation is needed.

Abstract: line 18 - Plural needed for limitations in the flwg sentence: “This review discusses some of the shortcomings and limitations of current noise regulations in the United States and Europe.”

Line 17 – remove “stress” – stress by itself is not life threatening. Revise to: “… such as cardiovascular disease and dementia”

Introduction

Line 36: Consider adding the word potentially and the following slight revision: “…potentially debilitating hearing-related impairments such as…”

Line 38-40: Please supply references for the assoc between NIHL and cognitive decline and social isolation. Fortunato et al (2016) is a review looking at the assoc between Hl and cognitive decline in ageing. One pop based study found NIHL to be associated with social isolation in older women, aged 60 to 69, but not older men (Mick et al. 2014). Chronic or long term exposure to environmental noise has been assoc with coronary heart disease (CHD) – it is not necessarily having NIHL that is assoc with CHD. According to Gan et al (2012), a 5 year exposure period with population based cohort study found that individuals residing in the highest noise decile had a 22% increase in CHD mortality compared to those in the lowest decile. This type of information could be added to Intro.

One study suggested that reducing environmental noise exposure might save lives by decreasing the prevalence of cardiovascular heart disease – Gan et al. (2012).

Fortunato, S., Forli, F., Guglielmi, V., De Corso, E., Paludetti, G., Berrettini, S., and Fetoni, A. R. (2016). “A review of new insights on the association between hearing loss and cognitive decline in ageing,” Acta Otorhinolaryngol. Ital. 36, 155–166.

Li, C. M., Zhang, X., Hoffman, H., Cotch, M. F., Themann, C. L., and Wilson, M. R. (2014). “Hearing impairment associated with depression in U.S. adults, National Health and Nutrition Examination Survey 2005–2010,” JAMA Otolaryngol. Head Neck Surg. 140, 293–302.

Mick, P. et al. 2014. The assoc between HL and social isolation in older adults. Otolaryngol.-Head & Neck Surg, Vol 150 (3), 378-384.

Gan WQ, Davies HW, Koehoorn M, Brauer M. Association of long-term exposure to community noise and traffic-related air pollution with coronary heart disease mortality. Am J Epidemiol 2012;175:898–906.

Carroll et al. (2017) Vital Signs: NIHL among adults – United States 2011-2012.

Page 1, line 39: As “debilitating” was used in line 36, I would recommend leaving it out here.

Page 1, line 38: Consider revising the sentence as follows: “Moreover, current regulations do not take into account co-morbidities associated with NIHL among older adults such as cognitive decline and social isolation, both of which may contribute to depression and anxiety in many individuals.” A population based study by Li et al (2014) reported an assoc between hearing impairment and depression among US adults. This reference could be added to the Intro.

Among individuals exposed to chronic environmental noise, such as traffic noise, there has been an increase in congestive heart failure mortality rates reported compared to those in low noise exposures (Gan et al. 2012). This reference could be added to the Intro.

Page 2, line 49: Reference: M Charles Liberman, Epstein et al. 2013. Please check if this is indicated accurately. Should it not be Liberman et al. (2013)?

Also, Page 2, line 48 – A. Sheppard, Liu …2018 and A.M. Sheppard et al. 2017.  The referencing seems inconsistent. In some cases, the initials are indicated at the beginning of the reference and in some cases they are not.

Page 2, line 53: Consider omitting “Here”. Revise the sentence. Something like: “In the following sections, current occup noise regulations will be reviewed. In addition, auditory and non-auditory consequences of occup noise that may negatively impact quality of life and long term health will be discussed”.

Page 2, line 62: the phrase “fail to address or fail to consider” is over-used in this manuscript. Consider other language such “do not address” or “omit discussion of” “do not take into consideration”.

Page 2, line 61: recommend omitting “but as we will discuss later” – this reads as a lecture and not a written publication, Change to: “…speech perception, but do not address other auditory and non-auditory deficits which may occur”.

Page 2, line 65: Change to: “Second is the spectral content, that is, noise levels most commonly measured using…. “ Or – instead of that is – use “which are”.  Either one. As it stands now, it is a disjointed sentence.

Line 72-73: “..at least until thresholds reach an asymptote at durations 72 > 24 h (Melnick & Maves, 1974).” This portion of the sentence may be incomprehensible to some or many readers. I would recommend re-writing and explaining in more simple language. If this can’t be done easily or simply, consider deleting this text – and the authors asking the question – is it really needed?

Page 4, line 153: Consider changing to: “… because they occur in conjunction with occup noise exposure…” or “…in addition to occupational noise exposures and may contribute to hearing loss”

Please add the word “may” – underlined above.

Lines 154-155: Split this into 2 sentences. New sentence – please revise as follows: “However, environmental noise is not accounted for when considering what constitutes hazardous noise levels with regards to hearing health”.

Page 4, line 177. Split into 2 sentences. Start 2nd sentence with: “However, standard clinical audiometric evaluations….”

Page 4, line 182: Change to: “.. exposed to hazardous levels of noise…”

Page 5, line 210: consider changing to: “..but show significantly lowered DPOAE amplitudes”. I am not sure that the term abnormal is the best to use in this context.

Page 5, line 213: I would say, “Indeed, several studies have indicated that.. or “.. an increasing number of studies…

Line 215-216: Change to: “For example, in a study of military personnel (n=76), findings indicated that those with prior noise exposure…..” Check the end of sentence – there is an extra period here.

Can you include another study here which has a larger sample size, to support this?

Page 5, line 203 – start new paragraph beginning with– “One assessment tool..

Page 5, line 213 – start new paragraph beginning with “Indeed, several studies have indicated…

Page 5, line 217 – revise and start new paragraph - “In addition, OAE test re-test reliability studies have shown less variance associated with this measure compared to audiometric pure-tone average thresholds” – Add at least 2 references here.

Page 5, line 220: End sentence with Hall & Lutman, 1999 reference. Start new sentence with:  “This represents a critical difference when thresholds are…”

Page 5, lines 223-224: change to: “…by the time pure-tone average threshold shifts would be detected during annual occupational hearing conservation testing”.

Page 5, line 226-227 - Add “hearing to: “..noise-induced hearing damage”

Page 5, line 227 – delete the word “now”.

Page 6, line 244 – use of hearing protection in occup settings around 38%. Daniell et al. 2006. Can you use a more recent reference? There is research showing that when HP is mandatory in an occup setting, the compliance is higher. See Feder et al (2017) – pop-based study showed 80% compliance when HP was mandatory but much less compliance when not mandatory. Also, studies showing compliance increases with noise level- see Danish pop based study – 75% of workers used HP when exposed to more than 85 dBA and 42% wore HP when exposed to 80 to 84 dBA (Rubak et al. 2006). Lumber mill workers – compliance dropped to 84% (from 100%) when noise was at or above 95 dBA and to 60% compliance when exposed to under 85 dBA (Davies, 2009).

Davies H. Occupational noise exposure and hearing protector use in Canadian lumber mills. J Occup Environ Hyg. 2009;6:32–41.

Feder et al. (2017). Prevalence of hazardous occcup noise exposure, hearing loss and hearing protection usage among a representative sample of working Canadians. JOEM, Vol 59 (1).

Page 6, lines 259-261: Split this sentence into 2 sentences. Revision suggestion indicated below.

End sentence with “…the most prevalent disability in the VA healthcare system.”  New sentence – revise as follows: “The high financial cost is borne out by estimates of VA disability payments totalling a staggering $2 billion U.S dollars (Durch et al. 2006; Yankaskas, 2013).

Page 6, lines 265-266: This sentence about animal models seems to come out of nowhere – no linkage to previous or succeeding sentence. What is the relationship to the topic? How does it relate to humans or what is the conjecture? Either remove it or expand to make a linkage to hyperacusis and as it may relate to the human experience in a meaningful way.

Page 6, line 271 – plural form of disorder needed – “… alongside other auditory disorders…”

Page 6, lines 272-274: More explanation is needed here. How is hyperacusis modified when the individual is exposed to stress or leisure noise? Does it increase or decrease? “Data from female pre-school teachers exposed to noise at work suggests a model of hyperacusis whereby the prevalence is modified by additional factors such as stress, annoyance or unrelated leisure noise (Fredriksson, Hussain-Alkhateeb, & Persson Waye, 2017).”

Page 6, line 277 and line 279: Change to: “.. many researchers believe that tinnitus emerges…   and that hyperacusis results from…”.

Page 6, lines 284 – perhaps the authors could add the exact tests or measures here –

“.. .undetectable by threshold measurements using audiometry or DPOAE testing.. “

Page 8, line 350: Change to: “… presented at noise levels of 90-100 dBA, increases the…”

Page 8, line 351: Consider changing to: “…of noise is novel to an individual’s environment…”

Page 8, line 378: Change to: “…exposed to occupational noise in addition to residential traffic noise would be at even greater risk for CVD”.

Page 8, line 381: Remove “and” after history of smoking. Replace with comma.

Page 9, line 392 – Place comma after 2009 - “In 2009, the cost of..”

Summary

Page 9, line 415: Should this be – “ in annual occupational hearing conservation programs” ? Or hearing conservation monitoring programs. Be consistent throughout manuscript with regards to this terminology.

Page 9, line 420: plural form of relationship needed – “3) establish dose-response relationships between…”

Another study showing DPOAEs to be more sentisitve than audiometry in determining subclinical cochlear damage, specifically in high freq OHC, manifested by decreased amplitudes or absent DPOAEs - Helleman HW, Jansen EJM, Dreschler WA. Otoacoustic emissions in a hearing conservation program: general applicability in longitudinal monitoring and the relation to changes in pure-tone thresholds. Int J Audiol.2010;49:410–419.

Author Response

Reviewer 1

Generally, I have no comments. This paper is well-elaborated. The literature data concerning auditory and non-auditory effects of occupational exposure to noise as the shortcomings and limitations of existing noise regulations are clearly and appropriately presented. Thus, I only suggest to implement some minor corrections/changes listed below.

We thank the reviewer for their detailed review and positive comments on the article.

  1. Lines 25-26
    Is: “… Leq: equivalent continuous sound level; LAeq; 8h: A-weighted equivalent continuous sound level; 8 hours; LDEN: day-evening-night level”

Should be:

Leq: equivalent continuous sound pressure level; LAeq,8h: A-weighted equivalent continuous sound pressure level; 8 hours; LDEN: day-evening-night level.”

Corrected

  1. Lines 74-78
    Is: “A common method of integrating both the noise intensity and duration is the equivalent continuous sound level (Leq), which represents the average noise level over a defined period. When Leq acoustic measurements are made for 8 h with the A-weighting scale, the term LAeq,8h is used to indicate the type of weighting scale and the duration of the exposures”

Should be:

“A common method of integrating both the noise intensity and duration is the equivalent continuous sound level (Leq), which represents the energy mean of the noise level averaged over a defined period. When Leq acoustic measurements are made for 8 h with the A-weighting filter, the term LAeq,8h is used to indicate the type of frequency weighting and the duration of the exposures”.

Corrected

  1. Line 82-83
    Is: “LDEN (dayevening-night level), which indicates the average A-weighted noise over a 24 h period”

Should be:

“LDEN (day-evening-night level), which indicates the average A-weighted sound pressure level over a 24 h period”,

Corrected

  1. Lines 131-135
    “Occupational environments are categorized into a lower action level if the 8-h TWA (denoted in the directive as LEX) reaches 80 dB A or a peak pressure value of 135 dB C. If noise reaches an 8-h TWA of 85 dB A or has a peak sound pressure value of 137 dB C, it is classified as an upper action limit level. Finally, the ceiling threshold for acceptable noise exposure is an 8-h TWA of 87 dB A or a peak sound pressure limit of 140 dB C” ”

Should be:

“Occupational environments are categorized into a lower action level if the LAeq,8h (called a daily noise exposure level and denoted in the directive as LEX,8h) reaches 80 dBA or a C-weighted peak sound pressure level of 135 dBC. If noise reaches a LAeq,8h of 85 dBA or has a C-weighted peak sound pressure level of 137 dBC, it is classified as an upper exposure action limit level. Finally, the ceiling threshold for acceptable noise exposure is an LAeq,8h of 87 dBA and a C-weighted peak sound pressure level of 140 dBC”.

Corrected

Reviewer 2 Report

Generally, I have no comments. This paper is well-elaborated. The literature data concerning auditory and non-auditory effects of occupational exposure to noise as  the shortcomings and limitations of existing noise regulations are clearly and appropriately presented. Thus, I only suggest to implement some minor corrections/changes listed below.

  • Lines 25-26
    Is: “… Leq: equivalent continuous sound level; LAeq; 8h: A-weighted equivalent continuous sound level; 8 hours; LDEN: day-evening-night level”

Should be:

Leq: equivalent continuous sound pressure level; LAeq,8h: A-weighted equivalent continuous sound pressure level; 8 hours; LDEN: day-evening-night level.”

  • Lines 74-78
    Is: “A common method of integrating both the noise intensity and duration is the equivalent continuous sound level (Leq), which represents the average noise level over a defined period. When Leq acoustic measurements are made for 8 h with the A-weighting scale, the term LAeq,8h is used to indicate the type of weighting scale and the duration of the exposures”

Should be:

“A common method of integrating both the noise intensity and duration is the equivalent continuous sound level (Leq), which represents the energy mean of the noise level averaged over a defined period. When Leq acoustic measurements are made for 8 h with the A-weighting filter, the term LAeq,8h is used to indicate the type of frequency weighting and the duration of the exposures”.

  • Line 82-83
    Is: “LDEN (dayevening-night level), which indicates the average A-weighted noise over a 24 h period”

Should be:

“LDEN (day-evening-night level), which indicates the average A-weighted sound pressure level over a 24 h period”,

  • Lines 131-135
    “Occupational environments are categorized into a lower action level if the 8-h TWA (denoted in the directive as LEX) reaches 80 dB A or a peak pressure value of 135 dB C. If noise reaches an 8-h TWA of 85 dB A or has a peak sound pressure value of 137 dB C, it is classified as an upper action limit level. Finally, the ceiling threshold for acceptable noise exposure is an 8-h TWA of 87 dB A or a peak sound pressure limit of 140 dB C” ”

Should be:

“Occupational environments are categorized into a lower action level if the LAeq,8h (called a daily noise exposure level and denoted in the directive as LEX,8h) reaches 80 dBA or a C-weighted peak sound pressure level of 135 dBC. If noise reaches a LAeq,8h of 85 dBA or has a C-weighted peak sound pressure level of 137 dBC, it is classified as an upper exposure action limit level. Finally, the ceiling threshold for acceptable noise exposure is an LAeq,8h of 87 dBA and a C-weighted peak sound pressure level of 140 dBC”.

Author Response

Reviewer 2

Title: Occupational Noise: Auditory and Non-Auditory Consequences (Review)

Manuscript ID: ijerph-993583

Comments and suggested revisions:

Overall, this is a well written and well laid out manuscript. It is topical and of great relevance. Very minor revisions indicated below. Would like to see a few more references added to the Intro and to the section on DPOAEs and hearing protection usage. There are a few places where more explanation is needed.

Thank you for your very detailed review of the article and positive comments. We very much appreciate the provided text edits and have incorporated most of the suggestion. In some instances, we felt the suggestions were more related to writing style and not necessarily beneficial to the articles clarity or impact and preferred the original style.

  1. Abstract: line 18 - Plural needed for limitations in the flwg sentence: “This review discusses some of the shortcomings and limitations of current noise regulations in the United States and Europe.”

Corrected

  1. Line 17 – remove “stress” – stress by itself is not life threatening. Revise to: “… such as cardiovascular disease and dementia”

Corrected

Introduction

  1. Line 36: Consider adding the word potentially and the following slight revision: “…potentially debilitating hearing-related impairments such as…”

Corrected

  1. Line 38-40: Please supply references for the assoc between NIHL and cognitive decline and social isolation. Fortunato et al (2016) is a review looking at the assoc between Hl and cognitive decline in ageing.

One pop based study found NIHL to be associated with social isolation in older women, aged 60 to 69, but not older men (Mick et al. 2014).

According to Gan et al (2012), a 5 year exposure period with population based cohort study found that individuals residing in the highest noise decile had a 22% increase in CHD mortality compared to those in the lowest decile. This type of information could be added to Intro. One study suggested that reducing environmental noise exposure might save lives by decreasing the prevalence of cardiovascular heart disease – Gan et al. (2012).

Fortunato, S., Forli, F., Guglielmi, V., De Corso, E., Paludetti, G., Berrettini, S., and Fetoni, A. R. (2016). “A review of new insights on the association between hearing loss and cognitive decline in ageing,” Acta Otorhinolaryngol. Ital. 36, 155–166.

Li, C. M., Zhang, X., Hoffman, H., Cotch, M. F., Themann, C. L., and Wilson, M. R. (2014). “Hearing impairment associated with depression in U.S. adults, National Health and Nutrition Examination Survey 2005–2010,” JAMA Otolaryngol. Head Neck Surg. 140, 293–302.

Mick, P. et al. 2014. The assoc between HL and social isolation in older adults. Otolaryngol.-Head & Neck Surg, Vol 150 (3), 378-384.

Gan WQ, Davies HW, Koehoorn M, Brauer M. Association of long-term exposure to community noise and traffic-related air pollution with coronary heart disease mortality. Am J Epidemiol 2012;175:898–906.

Carroll et al. (2017) Vital Signs: NIHL among adults – United States 2011-2012.

Thank you very much for the associated references, we have included the reference into the introduction.

  1. Page 1, line 39: As “debilitating” was used in line 36, I would recommend leaving it out here.

We left the word “debilitating” in to accommodate for Reviewer #1’s wording suggestion.

  1. Page 1, line 38: Consider revising the sentence as follows: “Moreover, current regulations do not take into account co-morbidities associated with NIHL among older adults such as cognitive decline and social isolation, both of which may contribute to depression and anxiety in many individuals.” A population based study by Li et al (2014) reported an assoc between hearing impairment and depression among US adults. This reference could

We have rephrased this sentence and included the suggested reference.

  1. Among individuals exposed to chronic environmental noise, such as traffic noise, there has been an increase in congestive heart failure mortality rates reported compared to those in low noise exposures (Gan et al. 2012). This reference could be added to the Intro.

We have added this reference to the introduction.

  1. Page 2, line 49: Reference: M Charles Liberman, Epstein et al. 2013. Please check if this is indicated accurately. Should it not be Liberman et al. (2013)?

  1. Also, Page 2, line 48 – A. Sheppard, Liu …2018 and A.M. Sheppard et al. 2017.  The referencing seems inconsistent. In some cases, the initials are indicated at the beginning of the reference and in some cases they are not.

Re: questions 8 and 9- We reformatted our references to the author-date style using Endnote’s cite while you write function. They should now all be consistently represented throughout the manuscript.

  1. Page 2, line 53: Consider omitting “Here”. Revise the sentence. Something like: “In the following sections, current occup noise regulations will be reviewed. In addition, auditory and non-auditory consequences of occup noise that may negatively impact quality of life and long term health will be discussed”.

Corrected

  1. Page 2, line 62: the phrase “fail to address or fail to consider” is over-used in this manuscript. Consider other language such “do not address” or “omit discussion of” “do not take into consideration”.

We have revised the text to reduce the frequency of the word “fail”

  1. Page 2, line 61: recommend omitting “but as we will discuss later” – this reads as a lecture and not a written publication, Change to: “…speech perception, but do not address other auditory and non-auditory deficits which may occur”.

Corrected

  1. Page 2, line 65: Change to: “Second is the spectral content, that is, noise levels most commonly measured using…. “ Or – instead of that is – use “which are”.  Either one. As it stands now, it is a disjointed sentence.

We have changed the grammar for this sentence.

  1. Line 72-73: “..at least until thresholds reach an asymptote at durations 72 > 24 h (Melnick & Maves, 1974).” This portion of the sentence may be incomprehensible to some or many readers. I would recommend re-writing and explaining in more simple language. If this can’t be done easily or simply, consider deleting this text – and the authors asking the question – is it really needed?

We do not find this statement incomprehensible, and find the current language used relatively simple. This seems more like a preferential change rather than one that would benefit the manuscript. The content and reference is actually critical to understanding noise induced threshold shifts and how it could relate to other environmental or leisure noise exposures. We left the text unchanged

  1. Page 4, line 153: Consider changing to: “… because they occur in conjunction with occup noise exposure…” or “…in addition to occupational noise exposures and may contribute to hearing loss”

Please add the word “may” – underlined above.

Unchanged – this is preference for writing style. The same point is made in the following sentence and we have made sure to include the word “may” in that sentence.

  1. Lines 154-155: Split this into 2 sentences. New sentence – please revise as follows: “However, environmental noise is not accounted for when considering what constitutes hazardous noise levels with regards to hearing health”.

Corrected

  1. Page 4, line 177. Split into 2 sentences. Start 2nd sentence with: “However, standard clinical audiometric evaluations….”

Corrected

  1. Page 4, line 182: Change to: “.. exposed to hazardous levels of noise…”

Corrected

  1. Page 5, line 210: consider changing to: “..but show significantly lowered DPOAE amplitudes”. I am not sure that the term abnormal is the best to use in this context.

Corrected

  1. Page 5, line 213: I would say, “Indeed, several studies have indicated that.. or “.. an increasing number of studies…

Corrected

  1. Line 215-216: Change to: “For example, in a study of military personnel (n=76), findings indicated that those with prior noise exposure…..” Check the end of sentence – there is an extra period here. Can you include another study here which has a larger sample size, to support this?

We edited the suggested text.

This is a relatively large human study, even likely above what a power analysis would yield necessary to find statistical significance, therefore we don’t feel a study with larger sample size is warranted.

  1. Page 5, line 203 – start new paragraph beginning with– “One assessment tool..

Unchanged – We believe a short preface of OHC function before discussing the OAE assessment tool reads better and can aid the understanding of more novice readers.

  1. Page 5, line 213 – start new paragraph beginning with “Indeed, several studies have indicated…

Corrected

  1. Page 5, line 217 – revise and start new paragraph - “In addition, OAE test re-test reliability studies have shown less variance associated with this measure compared to audiometric pure-tone average thresholds” – Add at least 2 references here.

Corrected

We added the following additional reference (Zhao and Stephens 1999)

Zhao, F. and D. Stephens (1999). "Test-retest variability of distortion-product otoacoustic emissions in human ears with normal hearing." Scandinavian Audiology 28(3): 171-178.

  1. Page 5, line 220: End sentence with Hall & Lutman, 1999 reference. Start new sentence with:  “This represents a critical difference when thresholds are…”

Unchanged – the suggested text is not grammatically correct.

  1. Page 5, lines 223-224: change to: “…by the time pure-tone average threshold shifts would be detected during annual occupational hearing conservation testing”.

Corrected

  1. Page 5, line 226-227 - Add “hearing to: “..noise-induced hearing damage”

This edit would insinuate that the person has functional deficit – this would be inappropriate to state when only referring to OAE measures. We have left his unchanged.

  1. Page 5, line 227 – delete the word “now”.

Already rephrased with other edits.

  1. Page 6, line 244 – use of hearing protection in occup settings around 38%. Daniell et al. 2006. Can you use a more recent reference? There is research showing that when HP is mandatory in an occup setting, the compliance is higher. See Feder et al (2017) – pop-based study showed 80% compliance when HP was mandatory but much less compliance when not mandatory. Also, studies showing compliance increases with noise level- see Danish pop based study – 75% of workers used HP when exposed to more than 85 dBA and 42% wore HP when exposed to 80 to 84 dBA (Rubak et al. 2006). Lumber mill workers – compliance dropped to 84% (from 100%) when noise was at or above 95 dBA and to 60% compliance when exposed to under 85 dBA (Davies, 2009).

Davies H. Occupational noise exposure and hearing protector use in Canadian lumber mills. J Occup Environ Hyg. 2009;6:32–41.

Feder et al. (2017). Prevalence of hazardous occcup noise exposure, hearing loss and hearing protection usage among a representative sample of working Canadians. JOEM, Vol 59 (1).

Thank you for the references. We have included them in this section

Page 6, lines 259-261: Split this sentence into 2 sentences. Revision suggestion indicated below.

End sentence with “…the most prevalent disability in the VA healthcare system.”  New sentence – revise as follows: “The high financial cost is borne out by estimates of VA disability payments totalling a staggering $2 billion U.S dollars (Durch et al. 2006; Yankaskas, 2013).

Corrected

  1. Page 6, lines 265-266: This sentence about animal models seems to come out of nowhere – no linkage to previous or succeeding sentence. What is the relationship to the topic? How does it relate to humans or what is the conjecture? Either remove it or expand to make a linkage to hyperacusis and as it may relate to the human experience in a meaningful way.

We have very little evidence in humans to know if noise exposure causes hyperacusis – however it has been seen in animals. For clarity we have added “(i.e., hyperacusis)” following “lower loudness discomfort levels”. The reference to animal models is important because it is currently the strongest evidence that suggests hyperacusis might occur from noise exposure.

  1. Page 6, line 271 – plural form of disorder needed – “… alongside other auditory disorders…”

Corrected

  1. Page 6, lines 272-274: More explanation is needed here. How is hyperacusis modified when the individual is exposed to stress or leisure noise? Does it increase or decrease? “Data from female pre-school teachers exposed to noise at work suggests a model of hyperacusis whereby the prevalence is modified by additional factors such as stress, annoyance or unrelated leisure noise (Fredriksson, Hussain-Alkhateeb, & Persson Waye, 2017).”

Corrected – it increases

  1. Page 6, line 277 and line 279: Change to: “.. many researchers believe that tinnitus emerges…   and that hyperacusis results from…”.

Corrected

  1. Page 6, lines 284 – perhaps the authors could add the exact tests or measures here –

“.. .undetectable by threshold measurements using audiometry or DPOAE testing.. “

We feel this would be a less accurate statement. For example, audiometry may not be abnormal in general but could be abnormal specifically in the high frequency range of hearing. We have left his unchanged.

  1. Page 8, line 350: Change to: “… presented at noise levels of 90-100 dBA, increases the…”

Corrected

  1. Page 8, line 351: Consider changing to: “…of noise is novel to an individual’s environment…”

Corrected

  1. Page 8, line 378: Change to: “…exposed to occupational noise in addition to residential traffic noise would be at even greater risk for CVD”.

Corrected

Page 8, line 381: Remove “and” after history of smoking. Replace with comma.

Corrected

Page 9, line 392 – Place comma after 2009 - “In 2009, the cost of..”

Corrected

Summary

Page 9, line 415: Should this be – “ in annual occupational hearing conservation programs” ? Or hearing conservation monitoring programs. Be consistent throughout manuscript with regards to this terminology.

Corrected to “hearing conservation monitoring programs”

Page 9, line 420: plural form of relationship needed – “3) establish dose-response relationships between…”

Corrected

Reviewer 3 Report

The structure of the paper doesn’t look like the review. It rather looks like a handbook chapter with a lot (even to much) of encyclopedic knowledge. There is no personal comments from authors. The reader doesn't know on what basis the data were chosen to be quoted. 

I can’t identify the people to read this paper. To say the truth, this knowledge is too detailed for laymen but well known to specialists.

I have no comments on the content that is correct but I do not like the form. This publication organizes the knowledge but does not bring the new insight, even the conclusions have been known for years.

Author Response

Reviewer 3

The structure of the paper doesn’t look like the review. It rather looks like a handbook chapter with a lot (even to much) of encyclopedic knowledge. There is no personal comments from authors. The reader doesn't know on what basis the data were chosen to be quoted. 

We are sorry the reviewer does not like the format of the review; however, it falls completely within the guidelines of the journal and both reviewer 1 and 2 praised the article for being “well-elaborated” and “well laid out”. Therefore, we don’t feel the paper needs to be re-organized. Possibly the reviewer was expecting a systematic review format?

I can’t identify the people to read this paper. To say the truth, this knowledge is too detailed for laymen but well known to specialists.

We disagree with this statement (and from the positive comments from reviewer 1 and 2 re: the clarity of the article we would assume they would as well).  We feel the article is comprehensive and written in clear simple language able to be comprehended by novices and experts alike.

I have no comments on the content that is correct but I do not like the form. This publication organizes the knowledge but does not bring the new insight, even the conclusions have been known for years.

We again disagree with this statement. Our article does offer new insights and conclusions by specifically outlining the deficiencies and shortcomings of current hearing test measures used in hearing conservation monitoring programs.
